# Effects of Potassium Application on Growth and Root Metabolism of *Salvia miltiorrhiza* under Drought Stress

Jingyu Li [1,†], Xiangui Mei [1,†], Jin Zhang [2], Zhenqiao Song [1], Shiqi Wang [1], Wensheng Chen [1], Xin Wei [1], Xinsheng Fang [1] and Jianhua Wang [1,*]

1   State Key Laboratory of Crop Biology, College of Agronomy, Shandong Agricultural University, Tai'an 271000, China; sdauljy@163.com (J.L.); meixiangui@163.com (X.M.); szqsdau@163.com (Z.S.); w9s5q17@163.com (S.W.); 13183287308@163.com (W.C.); weixin0632062@163.com (X.W.); xinshf@126.com (X.F.)
2   Taishan Academy of Forestry Sciences, Tai'an 271000, China; zhjin0214@163.com
*   Correspondence: sdauwangjh@163.com
†   These authors contributed equally to this work.

**Abstract:** Potassium application can effectively mitigate the effects of drought stress on plant growth, and few studies have reported its application to the medicinal plant *Salvia miltiorrhiza* (*S. miltiorrhiza*). Four experimental treatments were used, including a control (Z1K0), non-drought potassium application (Z1K3), drought treatments (Z2K0), and drought-stress potassium application (Z2K3). The findings revealed that, in contrast to Z2K0, Z2K3 promoted the absorption of potassium in *S. miltiorrhiza*, elicited the aggregation of osmoregulatory compounds such as soluble protein and proline, and mitigated membrane impairment as a defense against the deleterious consequences of drought stress. Additionally, we extended our investigation to encompass comprehensive metabolomics analysis of the roots. Interestingly, subsequent root metabolomics analyses demonstrated that the drought application of potassium not only significantly reduced the amino acid content, but also increased the amount of terpenoids and phenolic acids in the roots. Nonetheless, the application of a particular amount of K under moderate drought conditions promoted the growth and yield of *S. miltiorrhiza*, but proved to be detrimental to its active ingredients. Indeed, the findings of this study offer valuable insights and recommendations for the application of potassium to mitigate the impact of drought stress in *S. miltiorrhiza* and other medicinal plants.

**Keywords:** *Salvia miltiorrhiza*; potassium application; metabolomics; drought stress

## 1. Introduction

Salvia is indeed a highly diverse genus, known for its numerous species distributed across various regions [1,2]. Plants in this genus are widely grown in China in temperate regions such as Shandong, Henan, Sichuan, and Shaanxi, with more than 66,700 hectares used for their cultivation [3,4]. Among the representatives of the Salvia genus, *Salvia miltiorrhiza* (*S. miltiorrhiza*) holds a special place as a widely utilized and popular herb [5]. It possesses significant economic, social, and medicinal value [6–8], making it a plant of great importance. Indeed, *S. miltiorrhiza* has a long history of use in traditional Chinese medicine for treating various ailments such as cardiovascular and cerebrovascular diseases, as well as diabetes mellitus. This herb is known for its rich composition of bioactive compounds, with phenolic acids and ketones being the two most vital, present in *S. miltiorrhiza*. Phenolic acids have anti-lipid peroxidation and anti-thrombotic activities and play a massive role in the treatment of cardiovascular diseases, inflammatory diseases, and metabolic diseases [9,10]. Ketones have anxiolytic effects [11], such as cryptotanshinone, which also has anti-inflammatory and neuroprotective effects, and there is an increasing demand for their production [3,12,13]. With the escalating demand, cultivation has emerged as a formidable means to harness the resources of *S. miltiorrhiza*. However, akin to most medicinal crops,

*S. miltiorrhiza* is found primarily in undulating and mountainous terrains, where irrigation is scant and cumbersome. China's largest *S. miltiorrhiza*-producing area is in Shandong, and *S. miltiorrhiza* seedling transplantation mainly takes place in March–April. The planting process requires sufficient water and the application of a large amount of organic fertilizer [14]. Due to the uneven spatial and temporal distribution of water, *S. miltiorrhiza* is often adversely affected by drought stress during its cultivation and growth process. Consequently, it frequently confronts the pernicious repercussions of drought-induced stress during its growth cycle [15]. This unwelcome state of affairs significantly stifles the proliferation and yield of *S. miltiorrhiza*. Drought not only reduces the biological yield of *S. miltiorrhiza* and disrupts the average physiological growth of the plant period, but also has a profound impact on the formation of quality compounds, significantly impeding the growth and production of *S. miltiorrhiza*. Therefore, determining how to alleviate the adverse effects caused by drought stress becomes an urgent problem to be solved. Since *S. miltiorrhiza* grows in mountainous areas where water is not readily available, studies have shown that drought conditions supplemented with the appropriate application of fertilizers can increase the physiological resilience of plants to abiotic stresses; thus, providing resistance to a certain degree of drought then becomes a feasible approach [16–18].

Potassium, an essential mineral element [19,20], plays a crucial role in plant physiological processes and regulates $K^+$ transport. This regulation helps to reduce the effects of drought stress [21,22]. However, limited research has been conducted on potassium application at different moisture levels to alleviate drought stress in medicinal plants like *S. miltiorrhiza*. The effects of potassium on *S. miltiorrhiza* under drought conditions are still unclear. Therefore, this study investigated the effect of potassium application under different moisture conditions on alleviating drought stress in *S. miltiorrhiza* and explored the possible mechanisms behind it. In addition, an analysis of the effect of growth and root secondary metabolism, physiological secondary metabolism, and fertilizer response mechanisms was also carried out.

## 2. Materials and Methods

### 2.1. Plant Materials and Treatments

The *S. miltiorrhiza* "JuXian Danshen" from China was selected as the experimental material. The experiment was conducted in a greenhouse at the Garden of Medicinal Plants, Shandong Agricultural University (36°17′ N, 117°17′ E). The experiment was a potting test, using 5 gallon pots that were 31 cm high, with a 27 cm upper caliber and 22 cm lower caliber. The soil comprised peat soil and perlite at a ratio of 7:3. The trial was conducted on 28 March 2021, when *S. miltiorrhiza* seedlings were transplanted, with one *S. miltiorrhiza* plant per pot, for a total of 50 pots of *S. miltiorrhiza* per treatment, and a total of 200 pots. Before transplanting, the soil had 37.26 g/kg of organic matter, 274.65 mg/kg of alkaline dissolved nitrogen, 22.43 mg/kg of quick-acting phosphorus, 28.43 mg/kg of quick-acting potassium, and a pH of 6.1. Two soil moisture treatments were designed for this experiment: the normal soil moisture Z1 (60–70% of the maximum soil water-holding capacity) and the drought treatment Z2 (45–55% of the maximum soil water-holding capacity). Each soil moisture gradient was set up with two potassium applications: K0 (containing $KH_2PO_4$ 0 g/L) and K3 (containing $KH_2PO_4$ 6.110 g/L). At 30 days after transplanting, each treatment was watered with a sufficient nutrient solution of the corresponding potassium content (nutrient solution: $Ca(NO_3)_2$ 0.938 g/L, $(NH_4)_2SO_4$ 0.0924 g/L, $MgSO_4 \cdot 7H_2O$ 0.567 g/L). The saturated water content of each pot was recorded before the water control experiment and then weighed on an electronic scale at 6:00 PM each day to control the soil water content of each treatment and reach the target water content. Sampling was carried out on the 60th day after transplanting and every 30 days thereafter to assess soluble protein and proline content. This was carried out a total of five times, and harvesting was carried out on the 220th day after transplanting (3 November 2021).

### 2.2. Physiological Characteristics

At the time of harvesting, 10 plants of *S. miltiorrhiza* were selected with uniform growth in each treatment. The plant height and root length were measured with a straightedge ruler. In addition, root length, root diameter, and basal stem thickness were measured with vernier calipers, and leaf thickness was measured with a thickness gauge (YH-1).

$$Biological\ yield\ =\ above-ground\ dry\ weight\ +\ below-ground\ dry\ weight$$

$$Harvest\ index\ =\ root\ dry\ weight/biological\ yield$$

### 2.3. Relative Water Content of Leaves

Using the saturation drying method [23], each treatment was performed in triplicate with 3 plants for each replication. A total of 5 leaves of *S. miltiorrhiza* were taken from each plant as a replication of one treatment. The fresh weight (FW) of each treated *S. miltiorrhiza* leaf was taken and quickly weighed, then soaked in deionized water for 24 h. The saturation fresh weight (TW) was recorded, after which the leaves were dried at 60 °C to a constant weight and the dry weight (DW) was measured. The relative water content (RWC) was calculated according to the formula: RWC (%) = (FW − DW)/(TW − DW) × 100%.

### 2.4. Determination of Potassium Content

The plants were harvested in triplicate for each treatment, replicated with 3 *S. miltiorrhiza* plants each time. *S. miltiorrhiza* was divided into three parts: leaves, stems, and roots, and then bagged at 105 °C for 30 min and dried at 60 °C to a constant weight. Dried tissues were finely ground in a post-grinding machine (Grinder 935A, Dongguan, China), sieved through 60 mesh, extracted with hydrochloric acid, and decolorized with activated carbon. The potassium content was determined using a flame photometer F-200 (Shanghai Yuananalysis Instruments Co., Ltd., Shanghai, China).

### 2.5. Proline Determination

For each sampling period, three *S. miltiorrhiza* plants per treatment were taken for the proline content assays [24]. The free proline content was determined using the hydrated ninhydrin method; 0.5 g of the fresh leaf sample was taken from *S. miltiorrhiza* and 5 mL of 3% sulfosalicylic acid solution was added. The sample was then extracted with boiling water for 10 min, and filtered through filter paper. After cooling, 4 mL of toluene was added and the mixture was shaken for 30 s. The supernatant was centrifuged at 3000 rpm for 5 min. The upper layer was aspirated and measured using a spectrophotometer at 520 nm.

### 2.6. Soluble Protein Determination

For each sampling period, three *S. miltiorrhiza* plants per treatment were taken for the soluble protein content assay. The soluble protein content was determined via the Kormas Brilliant Blue G-250 staining method [25]. A total of 0.2 g of sample was weighed in a pre-chilled mortar, and 1.6 mL of 50 mmol/L pre-chilled phosphate buffer (pH 7.8) was added. This was then ground into a homogenate on an ice bath, transferred into a centrifuge tube, and centrifuged at 4 °C and 10,000 rpm for 20 min. The supernatant was the protein crude extract. A total of 100 µL of the enzyme solution was taken, 2.9 mL of Komas Brilliant Blue solution was added, and the reaction was carried out for 2 min and measured by spectrophotometer at 595 nm. The TSP unit was mg/g FW.

### 2.7. Determination of Active Ingredients

For each treatment, three root tissues of *S. miltiorrhiza* were taken and harvested, dried in a blast drying oven at 60 °C, dried to a constant weight, crushed with a low-temperature wall-breaking machine, and sieved through an 80-mesh sieve. A total of 1 g of sieved *S. miltiorrhiza* powder was weighed with precision and placed in a conical flask. Then, 35 mL

of 80% methanol solution was added with precision, and ultrasonication-assisted extraction was carried out for 30 min under conditions of 40 Hz, 200 W, and 25 °C. The extract was passed through a 0.22 μm filter, and stored at 4 °C until analysis. The active ingredient content of *S. miltiorrhiza* was determined using UltiMate 3000 (Thermo, Waltham, MA, USA). The specific parameters of the liquid phase were: 0.1% acetic acid in water (A) and 0.1% acetic acid in acetonitrile (B); gradient elution: 0~0.5 min, 10%~10% B; 0.5~7 min, 10%~100% B; 7~8.5 min, 100%~100% B; 8.5~8.6 min, 100%~10% B; 8.6~10 min, 10%~10% B; flow rate 0.3 mL/min; column temperature 35 °C; and injection volume 0.3 μL.

The controls of rosmarinic acid, salvianolic acid A, salvianolic acid B, cryptotanshinone, tanshinone I, and tanshinone IIA were purchased from the China Institute of Food and Drug Control. They were diluted to five different concentrations with 80% methanol to construct the equation of concentration (x, μg/mL) versus peak area (y). The regression equations are shown in Table S1. The concentration was calculated by comparing the sample's peak area with the standard's peak area.

### 2.8. Sample Preparation and Metabolite Extraction

At the time of harvesting, three root samples of *S. miltiorrhiza* with uniform growth were taken from each treatment for the *S. miltiorrhiza* root metabolomics assay. The obtained samples were frozen in liquid nitrogen and stored at −80 °C. Sample preparation, extraction analysis, metabolite identification, and quantification were performed at Wuhan MetWare Biotechnology Co., Ltd. (Wuhan, China) (www.metware.cn, processed on 17 November 2021). *S. miltiorrhiza* samples were first placed in a lyophilizer (Scientz-100F, Ningbo, China) for vacuum drying and ground into a powder using a grinder (NM400, Retsch, Haan, Germany). A total of 100 mg of the powder was dissolved in 1.2 mL of 70% methanol extract and vortexed every 30 min for 30 s, for a total of 6 times. The sample was then passed through a filter membrane and used in the UPLC-MS/MS system for analysis. The AB4500Q TRAPUPLC-MS/MS system was used. The analytical conditions were as follows: UPLC (SHIMADZU Nexera X2, http://www.shimadzu.com.cn/, accessed on 14 October 2023); Agilent SB-C18 column (1.8 μm, 2.1 mm × 100 mm); and mobile phase A (pure water containing 0.1% formic acid) and phase B (acetonitrile). Sample measurements were performed using the following gradient procedure: 0 min, 95:5 (V(A):V(B)), where phase A changed to 5% and phase B changed to 95% within 9 min and was held for 1 min; 10–11 min, phase A changed to 5% and equilibrated for 14 min for the next shot injection. The flow rate was 0.35 mL/min, the column temperature was 40 °C, and the injection volume was 4 μL.

### 2.9. Metabolite Characterization

Metabolite identification and quantification were performed at Wuhan MetWare Biotechnology Co., Ltd. Mass spectrometry data were processed using the software Analyst 1.6.3, and metabolite quantification was performed using multiple reaction monitoring (MRM) modes of a triple quadrupole mass spectrometer. The screening criteria for differentially accumulated metabolites (DAMs) were fold change $\geq 2$ or fold change $\leq 0.5$ and VIP (variable importance in projection) $\geq 1$. Since the variability of *S. miltiorrhiza*-specific differential metabolites in drought-applied potassium was less than 2, we set the screening criteria to fold change $\geq 1.2$ or fold change $\leq 0.83$ and VIP (variable importance in projection) $\geq 1$ for the study of *S. miltiorrhiza*-specific differential metabolites.

### 2.10. Data Analysis

Physiological data were analyzed via analysis of variance (ANOVA) using DPS 7.05 and the results were expressed as the mean and standard deviation. Duncan's test was performed to test the statistical significance of differences ($p < 0.05$) between the means. Histograms were produced using Origin 2022. The principal component analysis (PCA) was plotted using the built-in statistical prcomp function of R software (R version 4.1.1, https://www.r-project.org/, accessed on 14 October 2023); heat maps, pyramidal plots,

enrichment plots, variance enrichment score plots, and flowcharts were drawn using Chemdraw 22 and AI 2019.

## 3. Results

### 3.1. The Effect of Potassium Application at Different Moisture Levels on Plant Height, Basal Stem Thickness, Leaf Water Content, and Leaf Thickness of S. miltiorrhiza

Drought reduced plant height and leaf thickness, and the drought application of potassium significantly increased basal stem thickness, leaf thickness, and plant height leaf water content compared to drought conditions alone. Compared with the control (Z1K0), non-drought potassium application (Z1K3), drought treatments (Z2K0), and drought-stress potassium application (Z2K3), increased basal stem thickness by 17.30%, 1.92%, and 11.53%, respectively. Both potassium application and drought significantly affected leaf thickness. Under the same moisture conditions, potassium application increased leaf thickness by 43.42% compared to normal moisture, and drought application increased leaf thickness by 43.10% compared to drought treatment. Under the same water conditions, potassium application improved leaf water use efficiency. Compared to the control (Z1K0), the relative water content of drought treatments (Z2K0) decreased by 9.22% and the relative water content of drought-stress potassium application (Z2K3) was significantly higher than that of drought treatments (Z2K0), by 9.30% (Table 1).

**Table 1.** Effect of potassium application on the growth of *S. miltiorrhiza* under drought stress.

| Treatment | Average Diameter of Stem (cm) | Blade Thickness (mm) | Plant Height (cm) | Relative Water Content (%) |
|---|---|---|---|---|
| Z1K0 | 0.52 ± 0.02 c | 40.67 ± 1.69 b | 55.00 ± 1.37 c | 75.35 ± 1.21 b |
| Z1K3 | 0.61 ± 0.07 a | 58.33 ± 2.31 a | 75.33 ± 2.56 a | 83.00 ± 0.82 a |
| Z2K0 | 0.53 ± 0.01 bc | 41.23 ± 2.64 b | 48.33 ± 1.95 d | 70.53 ± 0.77 c |
| Z2K3 | 0.58 ± 0.06 ab | 59.00 ± 1.53 a | 62.17 ± 0.56 b | 77.09 ± 1.36 b |

Based on Duncan's test; means shown by same letters are non-significant at 5% level.

Drought significantly reduced *S. miltiorrhiza* plant height, and drought-stress potassium application (Z2K3) increased *S. miltiorrhiza* plant height by 28.64% compared to drought treatments (Z2K0). This indicates that potassium application under drought conditions helps to resist the adverse effects of drought stress.

### 3.2. Effect of Potassium Application at Different Moisture Levels on Osmoregulatory Substances of S. miltiorrhiza

Potassium application under different moisture conditions favored the accumulation of potassium ion content in roots, stems, and leaves, showing that leaves > stems > roots. This indicated that potassium application significantly increased the potassium content in plants. Compared with drought conditions, the potassium content in leaves increased by 3.24% (Figure 1A), the potassium content in stems increased by 14.57% (Figure 1B), and the potassium content in roots increased by 17.16% with drought application (Figure 1C). The potassium ion concentration in plants under drought stress conditions was significantly higher than under normal water conditions, and the plants absorbed more potassium for plant stress tolerance.

The proline level of *S. miltiorrhiza* showed a trend of increasing and then decreasing (Figure 1D), and the soluble protein level showed an increasing trend throughout the reproductive period (Figure 1E). Drought treatments (Z2K0) significantly increased proline and soluble protein contents. Compared with drought treatments (Z2K0), drought-stress potassium application (Z2K3) increased proline and soluble protein contents by 8.64% and 44.18%, respectively.

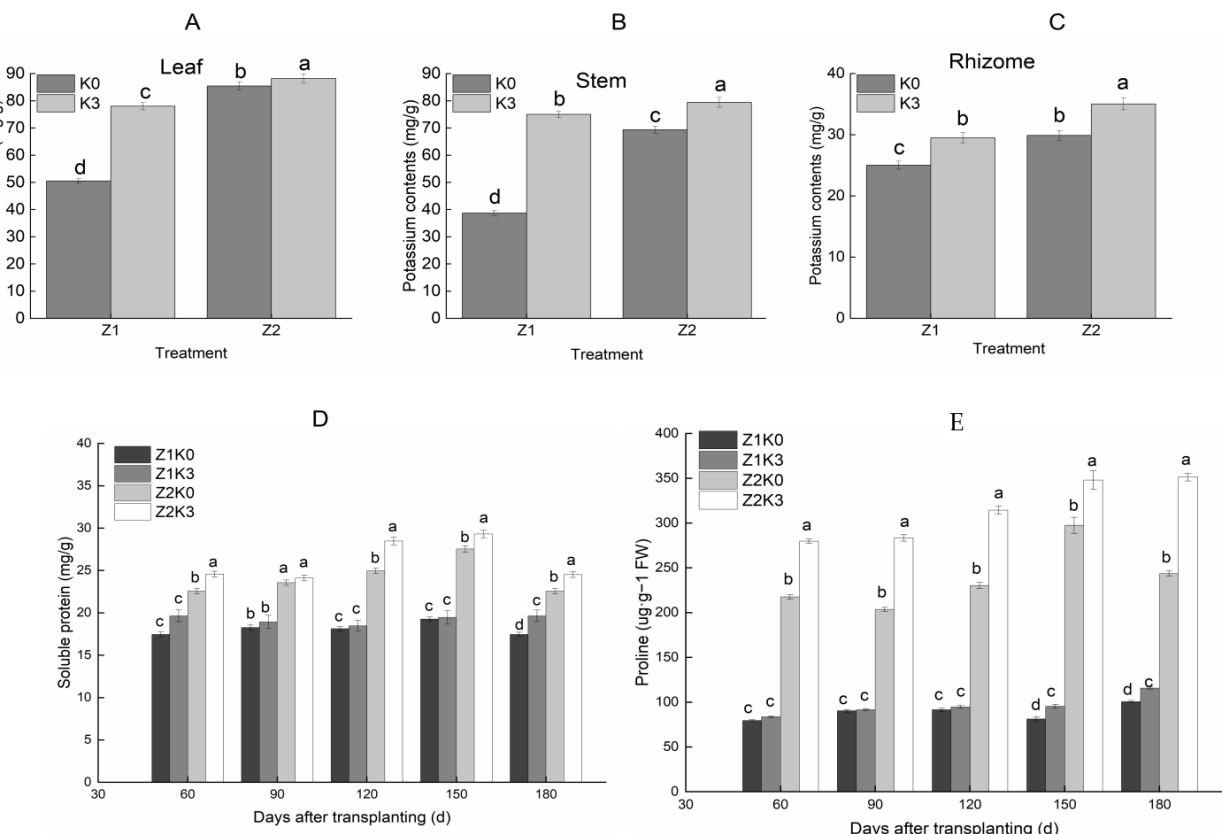

**Figure 1.** (**A–C**) Potassium content of different plant parts. (**D**) Effect of different moisture applications of potassium on proline content. (**E**) Effect of different moisture applications of potassium on soluble protein content. According to Duncan's test, letters above the error bars indicate a significant difference at $p < 0.05$.

### 3.3. Effect of Different Moisture Applications of Potassium on Yield and Its Traits in S. miltiorrhiza

Under the same moisture conditions, potassium application favored the main root length, main root thickness, and lateral root number of *S. miltiorrhiza*. The maximum value of each index was reached in the K3 treatment under normal moisture conditions. Compared with the control (Z1K0), the main root length, lateral root number, root dry weight, and biological yield decreased by 22.45%, 126.60%, 34.28%, and 44.82% under drought stress (Z2K0), respectively. Furthermore, after potassium application, the main root length, main root thickness, root dry weight, and biomass increased by 48.11%, 57.40%, 220.00%, 126.34%, and 101.53%, respectively. The harvest index was significantly higher after potassium application and showed K3 > K0 under different moisture conditions. A 10.71% higher harvest index was observed with drought application compared to drought conditions alone (Table 2 and Figure S1).

**Table 2.** Effect of different soil moisture application of potassium on yield traits and harvest index of *S. miltiorrhiza*.

| Treatment | Main Root Length (cm) | Main Root Width (cm) | Lateral Root Number | Root Yield (g/Plant) | Biomass (g/Plant) | Harvest Index |
|---|---|---|---|---|---|---|
| Z1K0 | 23.67 ± 3.79 b | 9.20 ± 1.16 b | 11.33 ± 1.15 b | 15.55 ± 1.18 c | 30.21 ± 1.92 c | 0.51 ± 0.02 b |
| Z1K3 | 38.00 ± 4.00 a | 16.24 ± 0.38 a | 18.33 ± 1.15 a | 32.85 ± 1.68 a | 51.48 ± 0.77 a | 0.64 ± 0.03 a |
| Z2K0 | 19.33 ± 1.15 b | 10.14 ± 1.57 ab | 5.00 ± 0.10 c | 11.58 ± 0.76 c | 20.86 ± 1.11 d | 0.56 ± 0.03 b |
| Z2K3 | 28.67 ± 7.64 ab | 15.96 ± 3.25 a | 16.00 ± 1.00 a | 26.21 ± 2.99 b | 42.04 ± 4.52 b | 0.62 ± 0.01 a |

Based on Duncan's test; means shown by same letters are non-significant at 5% level.

### 3.4. Metabolic Assays and Analysis of Potassium Application at Different Moisture Levels on S. miltiorrhiza

To investigate the changes in metabolites in the roots of *S. miltiorrhiza* with different moisture applications of potassium, we conducted metabolome assays in the roots of *S. miltiorrhiza*. A total of 801 metabolites were identified and divided into 11 major groups (Figures 2A and S3); the expression of metabolites of *S. miltiorrhiza* with three biological replicates of the control (Z1K0), drought treatments (Z2K0), non-drought potassium application (Z1K3), and drought-stress potassium application (Z2K3) under different conditions were subjected to principal component analysis. It was found that the samples of biological replicates were clustered together, indicating that the samples were well replicated and differed significantly among the groups (Figures 2B and S2).

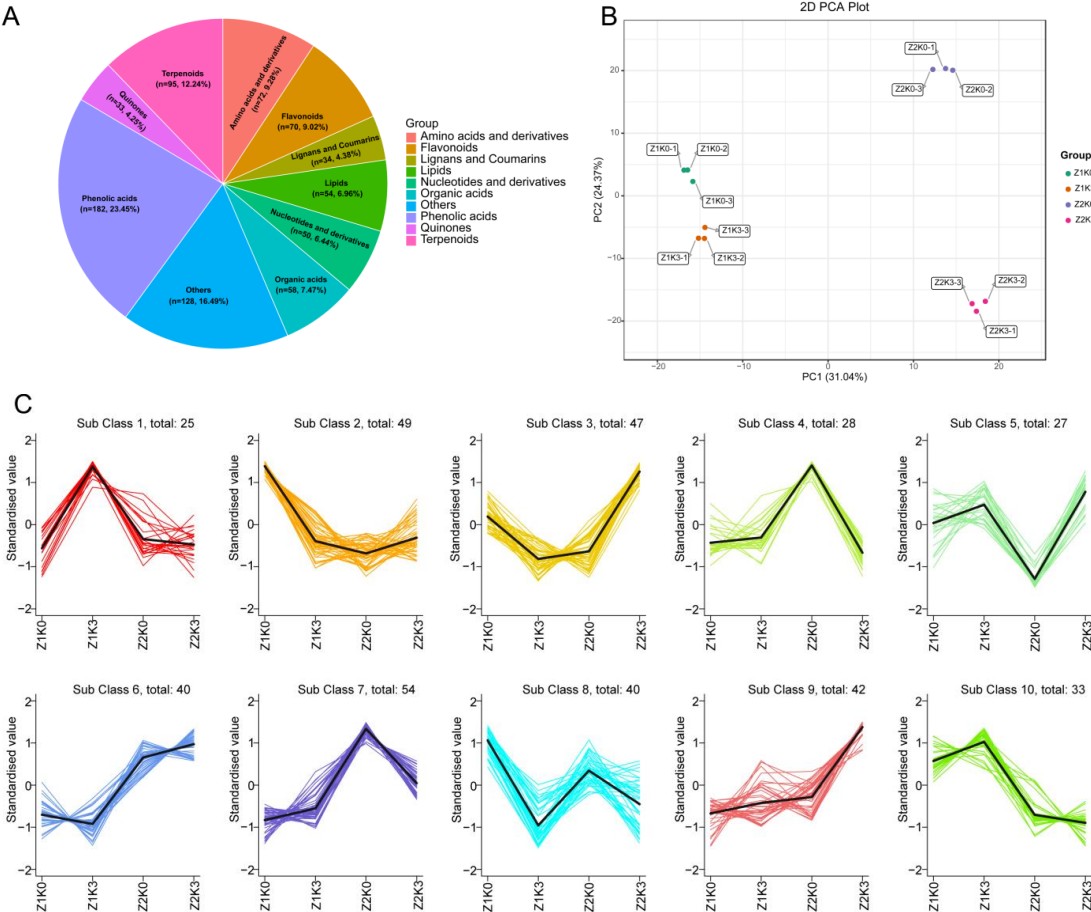

**Figure 2.** (**A**) Metabolite classification overview. (**B**) Sample principal component analysis. (**C**) K-means plot of differential metabolites, where subclass 5 and subclass 8 differential metabolites are associated with potassium application.

K-means analysis was used to standardize and center the relative contents of DAMs to study the trend in their relative contents in different samples. After subjecting four groups of samples to K-means analysis, we obtained 10 subclasses, as shown in Figure 2C. This experiment was followed to plot the categorical statistics (Table S3). The secondary metabolism of subclass 5 (Figure S4) and subclass 8 (Figure S5) was associated with the regulation of potassium application. Compared with the control (Z1K0), drought treatments (Z2K0) downregulated the top three classes of subclass 5 differential metabolites, namely phenolic acids, nucleotides and their derivatives, and organic acids, and then significantly upregulated them after the drought application of potassium; the differential metabolites in subclass 8 were the highest in the control (Z1K0) treatment, and drought downregulated

sugars and alcohols, indole alkaloids, and triterpenes compared with the control (Z1K0). They continued to be downregulated after drought-stress potassium application (Z2K3).

### 3.5. Differential Metabolite Screening

According to the statistics, the number of differential metabolites among the compared combinations is shown in Table S2. Among them, 39 DAMs were significantly upregulated, and 84 DAMs were significantly downregulated in the combination of Z1K0–Z1K3; 92 DAMs were significantly upregulated, and 67 DAMs were significantly downregulated in the combination of Z2K0–Z2K3. Drought-stress potassium application affects more secondary metabolites. Therefore, we will focus on the changes in secondary metabolites in *S. miltiorrhiza* roots under drought-stress potassium application.

In this study, the differential metabolites (Z1K0–Z1K3 and Z2K0–Z2K3) for different moisture applications of potassium were further analyzed using a clustering heat map analysis (Figure 3B,D). The differential metabolites were inconsistent among the different water application groups, with the Z1K0–Z1K3 normal water application being more associated with phenolic acids, triterpenoids, flavonoids, indoles, and alkaloids. Moreover, most of the metabolites were downregulated (Figure 3A,B). The differential substances in the Z2K0–Z2K3 combination predominantly belonged to the categories of phenolic acids, terpenoids, amino acids and their derivatives, nucleotides and their derivatives, and organic acids, sugars, and alcohols. Most of these substances were significantly upregulated (Figure 3C,D).

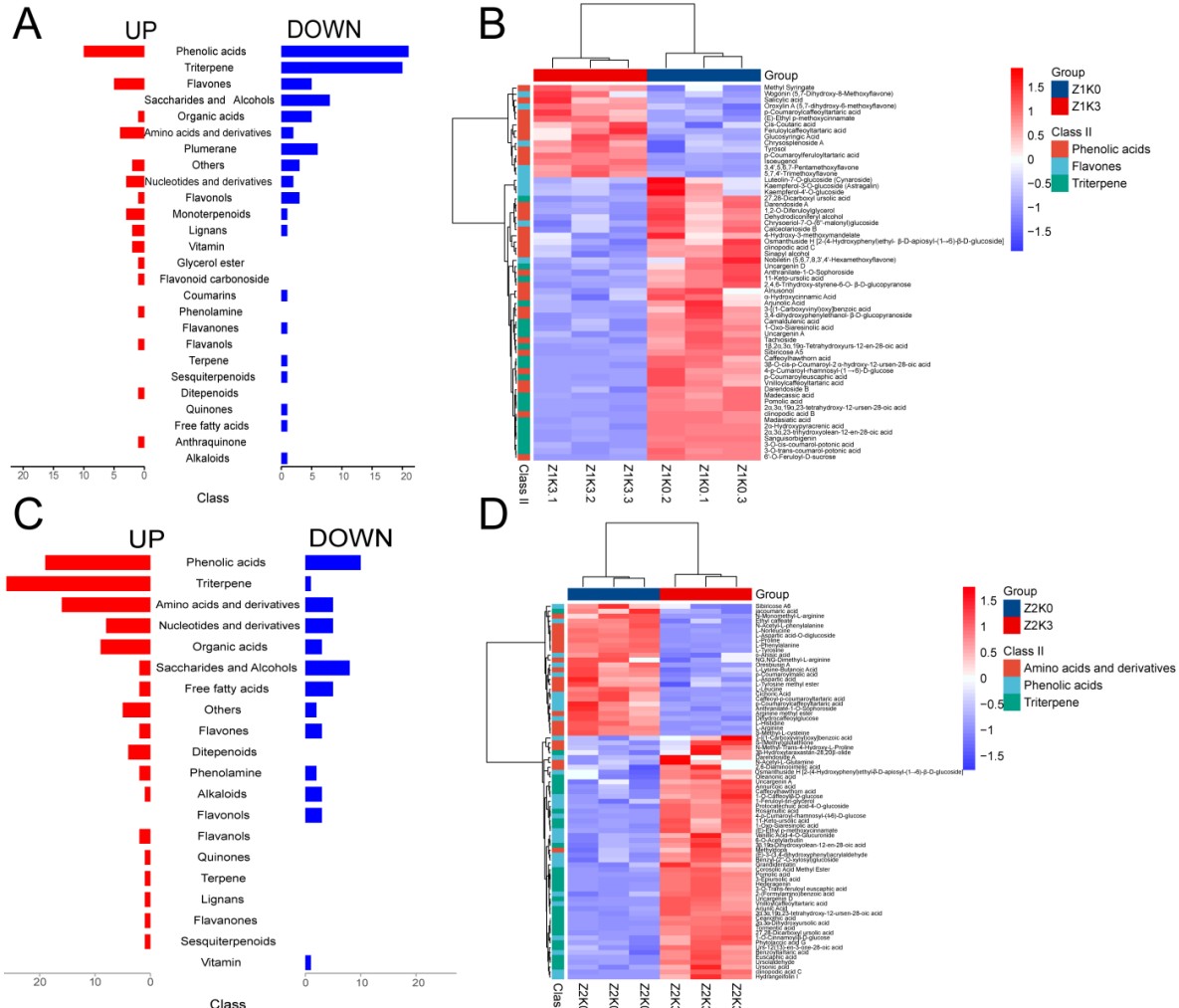

**Figure 3.** Classification of differential metabolites in Z1K0–Z1K3 and Z2K0–Z2K3 *S. miltiorrhiza* roots (**A**,**C**) and clustering heat map of the top three classes of differential metabolites (**B**,**D**).

In the Z2K0–Z2K3 combination, the DAMs were mainly enriched in the biosynthesis of secondary metabolites, ABC transporters, and amino acid biosynthesis pathways (Figure 4C,D). Compared with the drought treatments (Z2K0), drought-stress potassium application could produce more secondary metabolites in response to drought, especially in ABC transporter proteins and amino-acid-related metabolites such as L-valine, L-isoleucine, L-proline, L-arginine, L-histidine, L-phenylalanine, L-norleucine, L-leucine, L-aspartic acid, and L-tyrosine, which showed a downregulation trend.

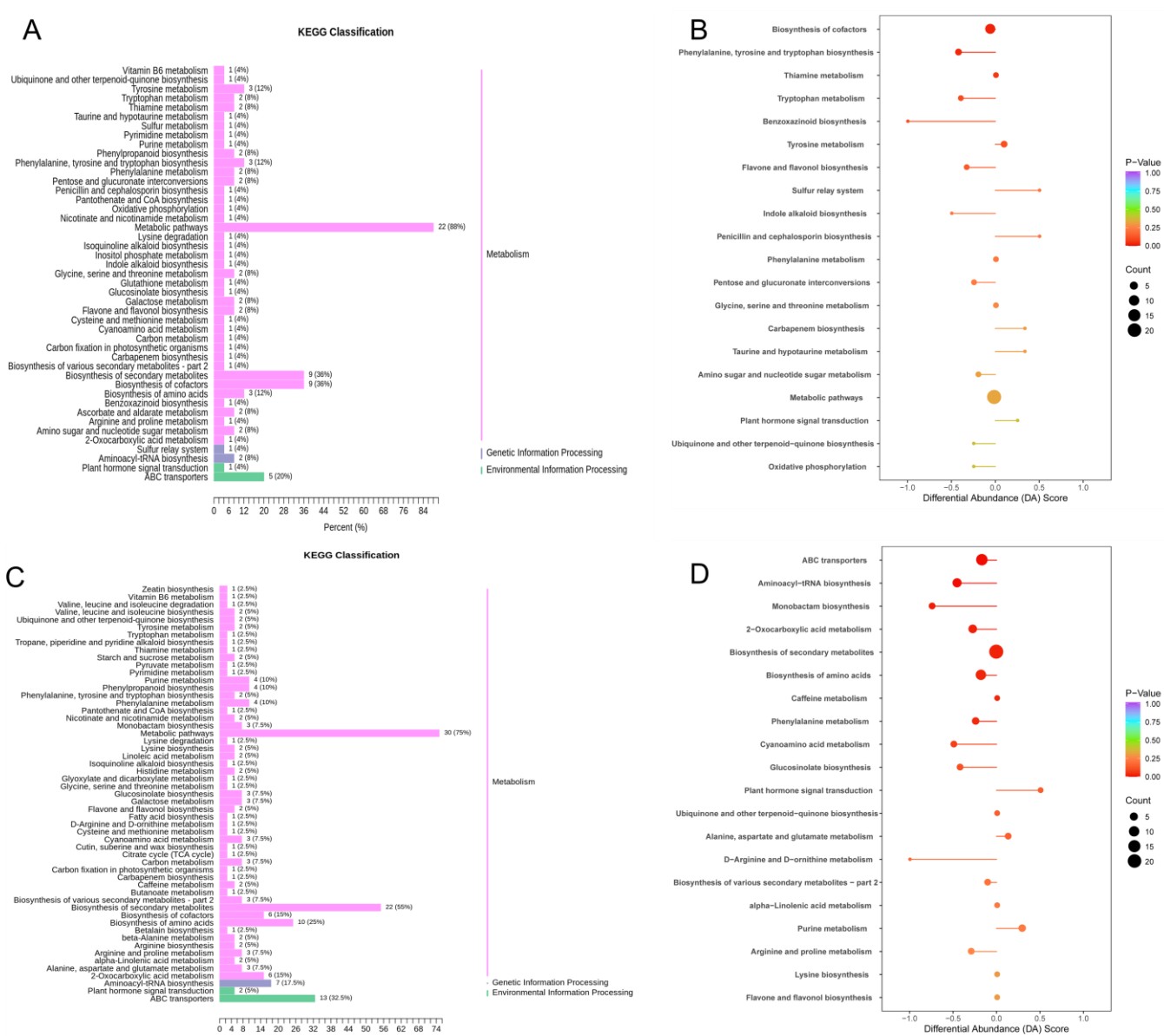

**Figure 4.** Z1K0–Z1K3 and Z2K0–Z2K3 differential metabolite KEGG classification plot (**A**,**C**) and differential abundance score plot (**B**,**D**).

In the two sets of comparisons, 30 overlapping differential metabolites were considered as key metabolites for the corresponding potassium application treatments, as shown in Figure 5A. To observe the changes between different metabolites more visually, their clustering heat map (Figure 5B) showed that phenolic acids and triterpenes decreased in Z1K0–Z1K3 with normal potassium application and increased in Z2K0–Z2K3 with drought application.

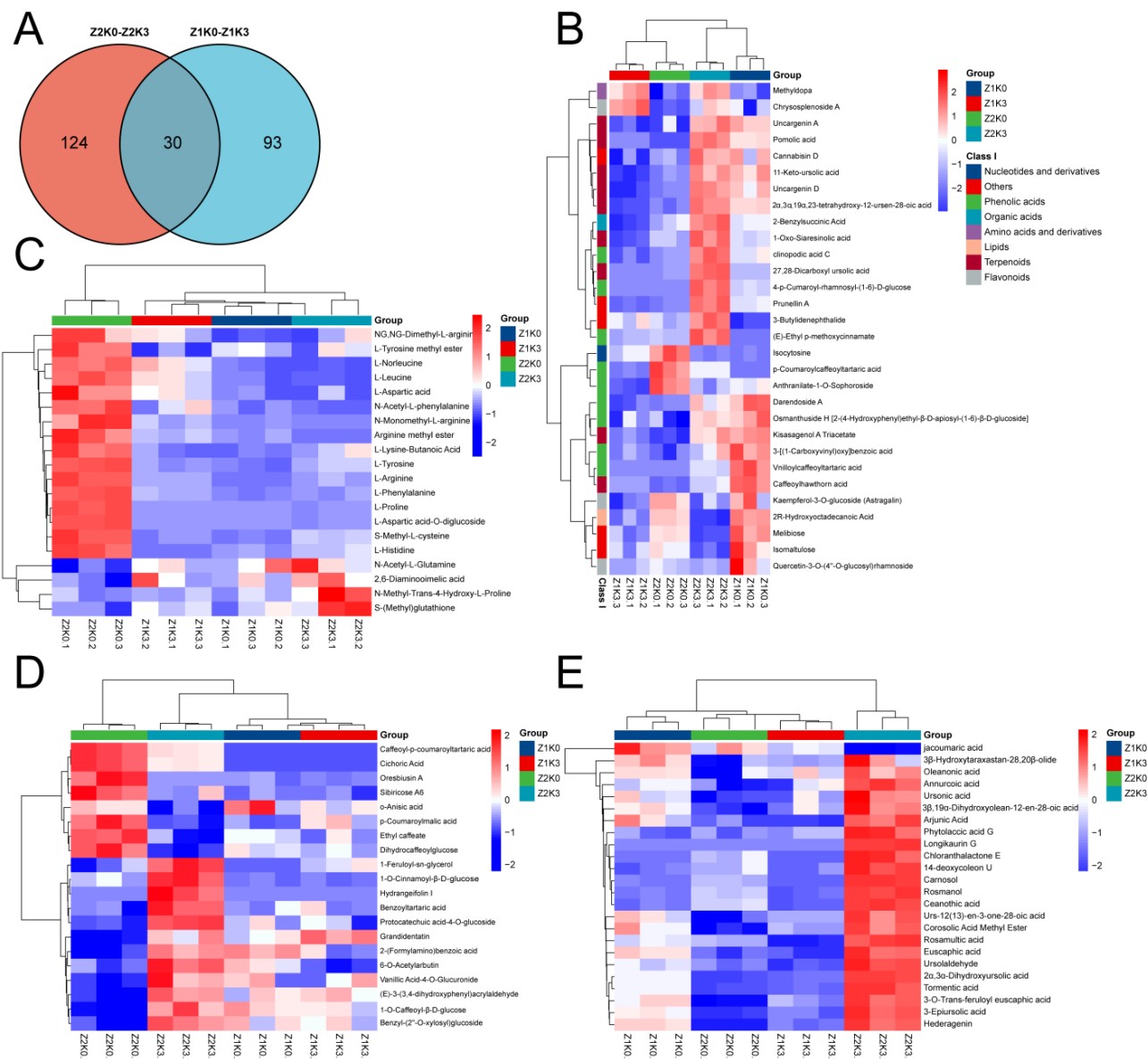

**Figure 5.** (**A**) Venn diagram of Z1K0–Z1K3 and Z2K0–Z2K3 differential metabolites. (**B**) Co-administered potassium differential metabolite heat map. (**C**) Z2K0–Z2K3-specific differential amino acid heat map. (**D**) Z2K0–Z2K3-specific differential phenolic acid heat map. (**E**) Z2K0–Z2K3-specific differential triterpenes heat map. The colors in the heat maps represent the normalized values corresponding to the horizontal rows.

Also, compared to non-drought potassium application (Z1K3), we were more concerned about the effect of the drought application of potassium, which affects more DAMs, as shown in Figure 5A. We categorized and analyzed the metabolites specific to the drought application of potassium (124) and found the most terpenoids (24), followed by phenolic acids (20) and amino acids and their derivatives (20). Analysis of phenolic acids revealed that both drought conditions (Z2K0 and Z2K3) enriched phenolic acids (Figure 5D). Also, drought-stress potassium application (Z2K3) increased triterpenoids compared to drought treatments (Z2K0), as shown in Figure 5E. These triterpenoids may be used for fungicidal disease resistance in plants and facilitate normal plant growth under drought conditions [26].

Drought more directly affected the primary metabolites. In Figure 5C, most amino acids and their derivatives were higher with the drought treatments (Z2K0) than with other treatments, After potassium application effectively reduced most amino acids such as

L-aspartate-O-diglucoside, L-proline, arginine methyl ester, N-monomethyl-L-arginine, N-acetyl-L-phenylalanine, L-arginine, L-histidine, L-lysine butyrate, L-phenylalanine, L-nor leucine, L-leucine, L-aspartic acid, S-methyl-L-cysteine, L-tyrosine, L-tyrosine methyl ester, and NG, NG-dimethyl-L-arginine was downregulated in the drought-stress potassium application (Z2K3). This study shows that potassium application under drought conditions reduces the accumulation of amino acids for physiological processes in *S. miltiorrhiza*.

### 3.6. Analysis of Metabolic Pathways of S. miltiorrhiza in Response to Water and Potassium Application

The results of metabolites in the roots of *S. miltiorrhiza* were integrated to map the metabolic pathways of *S. miltiorrhiza* in response to the water application of potassium based on metabolomics. As can be seen in Figure 6, metabolites in roots subjected to drought and potassium application showed different changes, with drought decreasing the content of most metabolites, including carbohydrates in roots. Compared to normal soil moisture, drought decreased carbon assimilation metabolites and elevated the metabolism of amino acids such as homoserine, threonine, saccharopine, arginosucinic and N-acetyl-lysine, 3-methylmalic, L-isoleucine, valine, alpha-isoproylmalate, pantothenate, and NAD+.

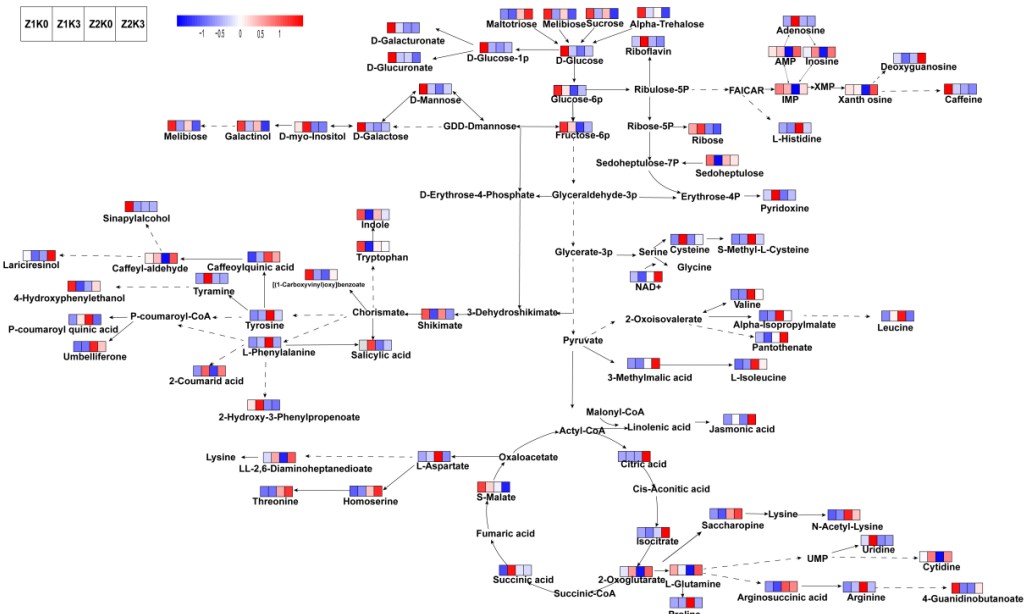

**Figure 6.** *S. miltiorrhiza* metabolic pathway.

Compared with the control (Z1K0), drought treatments (Z2K0) reduced the conversion of glucose, glucose 1 phosphate, D-glucose-1p, D-galacturonate, and D-glucuronate in gluconeogenesis. This reduced D-mannose, D-galactose, and galactinol in the gluconeogenic fructose-6p to melibiose process. Myo-inositol and galactinol content, while decreasing [(1-carboxyvinyl)oxy]benzoate, 2-coumarid acid, caffeyl-aldehyde, 4-hydroxyphenylethanol, sinapyl alcohol, and lariciresinol, elevated tyrosine, caffeoylquinic acid, L-phenylalanine, P-coumaroyl quinic acid, and umbelliferone. This resulted in the simultaneous elevation of the aspartate metabolic processes involving threonine and homoserine.

Compared to the control (Z1K0), non-drought potassium application (Z1K3), which mainly affected the mangiferylic acid branching acid pathway, decreased shikimate, salicylic acid, tryptophan, indole, [(1-carboxyvinyl)oxy]benzoate, 2-hydroxy-3 phenylpropenoate, vanillylmandelic acid, 4-hydroxyphenylethanol, and sinapyl alcohol.

Compared to the drought treatments (Z2K0), the drought-stress potassium application (Z2K3) upregulated amino acid nucleotide metabolism in adenosine monophosphate (AMP), inosincacid (IMP), inosine, xanthosine, deoxyguanosine caffeine, and L-Histidine. Drought-applied potassium increased the tricarboxylic acid cycle in citric acid, isocitrate,

and 2-oxoglutarate content, which improves plant immunity and related resistance, and organizes cell aging apoptosis to improve plant growth potential.

### 3.7. Effect of Moisture and Potassium Application Treatments on the Synthesis of Salvianolic Acid and Tanshinone

In this experiment, in order to verify the effect of drought-stress potassium application on the metabolite changes in the synthesis pathway of *S. miltiorrhiza* phenolic acid and tanshinone (screening condition: fold change $\geq$ 1.2 or fold change $\leq$ 0.83), according to the clustering heat map (Figure S6), α-(3,4-dihydroxyphenyl)lactic acid, L-phenylalanine, L-tyrosine, salvianolic acid C, lithospermic acid, methyl salvianolate H, salvianolic acid G, salvianolic acid A, salvianolic acid N, and salvianolic acid B were significantly enriched in the drought treatment (Z2K0).

Compared with the drought treatment, drought-stress potassium application led to α-(3,4-dihydroxyphenyl)lactic acid, L-phenylalanine, L-tyrosine, salvianolic acid C, lithospermic acid, salvianolic acid G, salvianolic acid A, salvianolic acid N, and salvianolic acid B decreased, but rosmarinic acid, clinopodic acid I, and salvianolic acid K were upregulated. This also caused isosalvianolic acid B, lithospermic acid B, salvianolic acid I, and salvianolic acid L contents to be lower than those of the treatment groups.

From the salvianolic acid synthesis pathway, this study found that drought positively affected salvianolic acid. Moreover, L-tyrosine, L-phenylalanine, tannins, salvianolic acid A, salvianolic acid C, and salvianolic acid B were synthesized in large amounts under drought conditions. In contrast, rosmarinic acid content was downregulated (Figure 7A). We paid more attention to the changes in the metabolic pathway of salvianolic acid after drought-stress potassium application. In this study, we found that caffeic acid and rosmarinic acid contents were enriched after drought-stress potassium application. Furthermore, drought-stress potassium application could increase the content of caffeic acid and decrease the content of salvianolic acid F, salvianolic acid A, and salvianolic acid B.

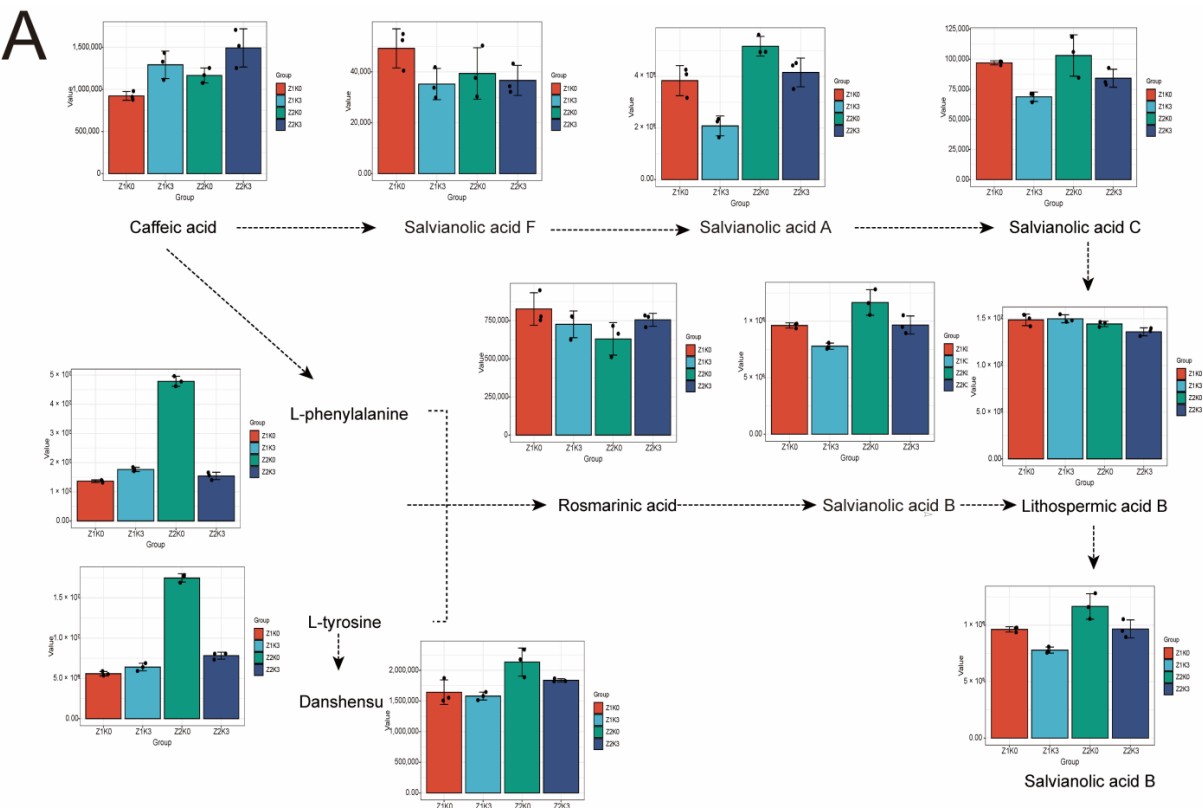

**Figure 7.** *Cont.*

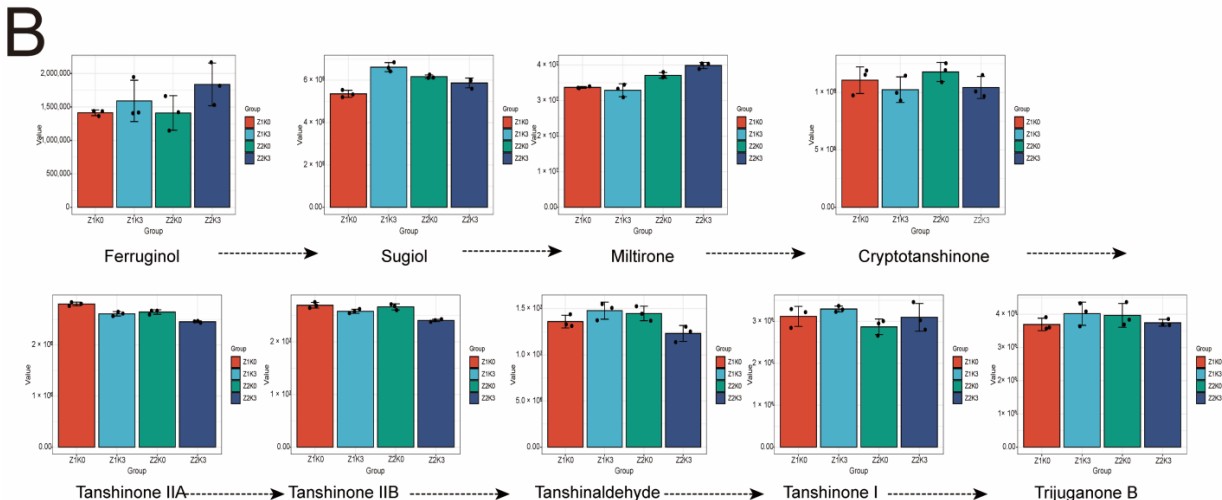

**Figure 7.** (**A**) Metabolite changes in the salvianolic acid pathway. (**B**) Metabolite changes in the metabolic pathway of tanshinone.

From the clustering heat map (Figure S6), it can be concluded that ferruginol, hypotannin quinone, rhamnol, dehydromiltirone, neocryptotanshinone, cryptotanshinone, tanshinone C, and 5-hydroxy-4′,6,7-trimethoxyflavone (salvigenin) were significantly upregulated after drought. In addition, tanshindiol B, tanshindiol C, trijuganone C, dihydroisotanshinone I, miltipolone, sugiol, 15,16-dihydrotanshinone I, and neoprzewaquinone A were downregulated after drought and increased through drought-stress potassium application. There was also a class of ketones, including tanshinone I, trijuganone B, tanshinaldehyde, 1-ketoisocryptotanshinone, methyltanshinonate, przewa tanshinone A, tanshinone IIB methylene tanshinquinone, 1,2-dihydrotanshinone I, and tanshinone IIA, that decreased after potassium application.

In this study, the effect of drought on tanshinone synthesis was not significant (Figure 7B). In the early stage of tanshinone synthesis, drought-stress potassium application was favorable to the accumulation of ferruginol and miltipolone contents, and drought-stress potassium application was not favorable to the accumulation of tanshinone IIA, tanshinone IIB, tanshinaldehyde, tanshinone I, and trijuganone B contents.

In this experiment, we quantified the main active ingredients of *S. miltiorrhiza* and found that drought significantly increased rosmarinic acid and salvia acid B content, but decreased tanshinone IIA content in *S. miltiorrhiza*. Compared with the drought treatments (Z2K0), drought-stress potassium application (Z2K3) significantly increased rosmarinic acid by 48.67% and tanshinone I by 8.09%, and decreased salvia acid B by 27.30% (Table 3). In addition, tanshinone B content decreased after drought application of potassium, which is basically consistent with the results of the metabolic assay above.

**Table 3.** Main active ingredient content of *S. miltiorrhiza*.

| Active Constituents (%) | Treatment | | | |
|---|---|---|---|---|
| | **Z1K0** | **Z1K3** | **Z2K0** | **Z2K3** |
| Rosmarinic acid | 0.0528 ± 0.0012 d | 0.1341 ± 0.0019 a | 0.0867 ± 0.0019 c | 0.1289 ± 0.0027 b |
| Salvia acid A | 0.0595 ± 0.0053 a | 0.0366 ± 0.0018 c | 0.0503 ± 0.0026 bc | 0.0458 ± 0.0026 ab |
| Salvia acid B | 3.53 ± 0.24 ab | 1.98 ± 0.09 c | 3.87 ± 0.04 a | 3.04 ± 0.55 b |
| Tanshinone I | 0.2009 ± 0.0101 b | 0.2301 ± 0.0037 a | 0.2114 ± 0.0055 b | 0.2285 ± 0.0020 a |
| Tanshinone IIA | 0.2112 ± 0.0091 a | 0.1859 ± 0.0148 a | 0.1220 ± 0.0041 b | 0.1229 ± 0.0181 b |
| Cryptotanshinone | 1.5732 ± 0.0068 a | 1.2781 ± 0.0001 b | 1.4537 ± 1262 a | 1.4025 ± 0.0440 a |

Based on Duncan's test, means shown by same letters are non-significant at 5% level.

## 4. Discussion

Moisture stress is the main abiotic factor affecting crop productivity, which impacts plant growth and yield [27]. Drought stress disrupts plant cell membranes and nuclei, and affects the plant water balance, which inhibits leaf and root growth [28]. In this study, we investigated the effect of potassium application at different moisture levels on the physiological characteristics of *S. miltiorrhiza*. Under the same moisture conditions, potassium application favored the increase in potassium ion concentration in all parts of *S. miltiorrhiza* and the growth of root indices [29]. Plants absorb potassium ions from the soil through their roots, and within a certain range, potassium uptake by plants increases with an increasing potassium concentration in the soil [30,31]. As in this experiment, K3 rootstock leaf potassium ion concentration treatment was higher than K1 treatment under the same water conditions. The increase in leaf potassium ions facilitates the use of ions in the plant for RUBP carboxylase activity, regulates stomatal opening, etc., and facilitates photosynthetic rate and photosynthetic product formation [32]. In the present study, potassium ion concentrations were much higher in the leaves and stems than in the roots. This suggests that $K^+$ uptake by plants after drought, which is used to regulate $K^+$ homeostasis in plant cells [20,33] and facilitate the repair and recovery of transport channels affected by previous drought, could effectively mediate the reopening of water-deficient stomata and the restoration of photosynthesis in *S. miltiorrhiza*, which is crucial for maintaining leaf water content [34].

Potassium fertilization at different water levels significantly increases proline and soluble protein contents. These osmoregulatory substances can maintain the regulatory role of cell expansion pressure on certain physiological functions, protect photosynthetic organs, and maintain photosynthesis [17,35]. Potassium application is beneficial to the vigorous growth of *S. miltiorrhiza*, which produces more dry matter and helps maintain the normal growth of *S. miltiorrhiza* during drought. In this study, the drought application of potassium facilitated the accumulation of proline and soluble protein, which are osmoregulatory substances used to protect leaf tissues from drought or to reduce the induced oxidative stress.

From the previous discussion, it should be clear that the moderate application of potash fertilizer is beneficial for crop substance yield [36–39]. Supplemental potassium fertilization is often required to achieve or maintain maximum crop yield, especially when the effective potassium content of the soil is low. There is evidence that potassium ion transport is associated with cell swelling, membrane transport, growth hormone homeostasis, cell signaling, and phloem transport [40–42]. $K^+$ is an important conventional regulator of plant root growth, and potassium application treatments have been shown to significantly increase the aboveground biomass and improve plant metabolism [43,44]. Moreover, the moderate application of K induced root development, increased root starch content, and consequently increased yield [37,45,46]. Some studies have further shown that high concentrations of potassium ions regulate plant root formation and growth by participating in the regulation of the growth hormone cytokinin. This further improves root dry matter accumulation [47] for drought tolerance or stress avoidance. In addition, potassium uptake is a key mechanism for plant stress response, so suitable concentrations of potash fertilizers under drought conditions are beneficial to the plant to produce positive effects [39,46,48,49]. In this study, potassium application at different moisture levels significantly improved the yield traits in the roots of *S. miltiorrhiza*, which was favorable for the increase in *S. miltiorrhiza*. In particular, when potassium was applied during drought, the yield of *S. miltiorrhiza* significantly increased.

The pathways of secondary metabolites of *S. miltiorrhiza* and their relationship to potassium application under different soil moisture conditions were systematically described using extensive targeted metabolomics analysis. The results showed that potassium application at different moisture levels induced complex changes in Salvia roots, including the TCA cycle, glycolysis/gluconeogenesis, amino acid metabolism, and phenylpropane metabolic pathways. The K-means analysis showed that, compared to drought treatments

(Z2K0), drought-stress potassium application (Z2K3) positively upregulated phenolics, nucleotides and their derivatives, and organic acids. Furthermore, drought-applied potassium treatment negatively downregulated sugars and alcohols, indole alkaloids, and triterpenes (Figures 2C, S4 and S5).

Drought stress affects the accumulation of key secondary metabolites (including phenolics and flavonoids). In addition, the increased accumulation of secondary metabolites prevents the oxidation of membrane structures via drought stress and confers greater pharmacological properties to the plant [50]. In this study, with the normal water application of potassium, the concentration of phenolic acids, triterpenoids, flavonoids, indoles, and alkaloids as secondary metabolites decreased (Figure 5A). Moreover, the drought application of potassium affected differential substances belonging to phenolic acids, triterpenes, and amino acids. Their derivatives, nucleotides and their derivatives, and organic acids, sugars, and alcohols were mostly upregulated (Figure 5C). Phenolic compounds have been shown to be major contributors to scavenging reactive oxygen species [51]. A large number of phenolics in this study were enhanced by drought-applied potassium and were further used to scavenge the adverse effects of drought. ABC transporter proteins are essential for plant development, and their associated ABC transporter protein genes (mainly in the roots) play an important role in drought tolerance [52,53]. Most of the metabolites in this study's ABC transporter protein family were upregulated in the drought treatment. Most of the differential metabolites in the enrichment pathway were downregulated after drought application of potassium. These results suggest that the drought application of potassium improved the metabolite pathway of *S. miltiorrhiza* under drought stress and was used more in the primary metabolism of *S. miltiorrhiza*.

Plants accumulate multiple metabolites in response to drought stress, including branched-chain amino acids (BCAAs) [54], and amino acid and carbohydrate metabolism play important roles in biological cellular adaptation [55]. Carbohydrate metabolism is coordinated with amino acid degradation, thus providing the carbon skeleton for the tricarboxylic acid cycle. This coordination may help plants to maintain energy balance during drought acclimation and facilitate recovery after stress mitigation. Under the drought treatments (Z2K0) in this study (Figure 5C), amino acids accumulated in large quantities. These osmoregulatory substances were induced by drought to cope with suboptimal conditions such as water deficit, probably mediated through the TCA cycle (the central metabolic cycle of most organisms). In contrast, drought-applied potassium reduced the amino acid content due to the ability of both amino acids and $K^+$ to mitigate drought stress [56,57]. The tricarboxylic acid cycle is enhanced through external potassium application, which leads to photosynthetic and carbohydrate-biased redistribution [58].

Compared to the drought treatments (Z2K0), drought-stress potassium application mainly affected amino acid nucleotide metabolism (Figure 4D), where the related metabolites AMP, IMP, inosine, xanthosine, deoxyguanosine caffeine, and L-histidine were significantly increased after drought-stress potassium application (Figure 5C). This is because histidine is beneficial to *S. miltiorrhiza* in regulating leaf stomata, resisting disease invasion and promoting the synthesis of cytokinins [29].

It has been shown that carbohydrates generate phenylalanine and lysine through the mangiferous acid pathway to synthesize salvianolic acid [59], while it has been shown that salvianolic acid can be used to enhance osmotic tolerance in plants [60]. In the present study, L-tyrosine, L-phenylalanine, tannins, salvianolic acid A, salvianolic acid C, and salvianolic acid B were synthesized in large quantities under drought conditions, which contribute to improving drought tolerance in *S. miltiorrhiza* (Figure S6). The drought-stress potassium application reduced the content of salvianolic acid F, salvianolic acid A, and salvianolic acid B (Figures S6 and 7). It is possible that potassium application and salvianolic acid substances for drought resistance both changed plant permeability. A similar pattern was observed for ketones in this experiment, where tanshinones were inversely proportional to root weight; as root weight increased, tanshinone content relatively decreased (Figures S7 and 7B), but

the accumulation of the total active ingredient content of a single *S. miltiorrhiza* plant was still favored.

## 5. Conclusions

In summary, potassium application at different moisture levels was beneficial to the yield of *S. miltiorrhiza*, especially in terms of main root length, main root thickness, and lateral root number. Compared with drought conditions, the drought application of potassium facilitated the enhancement of osmotic material and reduced drought stress injury in *S. miltiorrhiza*. In this experiment, drought conditions were found to significantly increase amino acid biosynthesis compared with non-drought conditions. However, when potassium was applied during drought, it was observed to decrease amino acid biosynthesis while simultaneously increasing the production of phenolic acids and terpenoids. This change in secondary metabolite production, characterized by an increase in phenolic acids and terpenoids, can contribute to improved plant disease resistance and facilitate the growth of *S miltiorrhiza* in adverse conditions. Under moderate drought conditions, a certain amount of potassium fertilizer was applied afterward, which was beneficial to the yield but not to the accumulation of most of the active ingredients of *S. miltiorrhiza*. These results suggest that the drought application of potassium plays an important role in alleviating abiotic stresses and that this alleviation is achieved by altering osmoregulatory substances and amino acids in *S. miltiorrhiza* after potassium application.

**Supplementary Materials:** The following supporting information can be downloaded at: https://www.mdpi.com/article/10.3390/agronomy13112796/s1, Figure S1. Status of *S. miltiorrhiza* at harvesting; Figure S2. (A) Mass spectral data PCA scores of each group of samples and QC samples. (B) Overall sample PC1 control chart. (C) Correlation of QC samples; Figure S3. Overall clustering chart of samples; Figure S4. Subclass 5 top 3 classes of differential metabolites; Figure S5. Subclass 8's top 3 classes of differential metabolites; Figure S6. Heat map of differential metabolites associated with salvianolic acid; Figure S7. Heat map of differentially metabolized substances associated with tanshinone; Table S1. Calibration curves of six standard chemicals. Table S2. Statistical table of the number of differential metabolites; Table S3. Secondary classification statistics of subclass differential metabolites for each experimental treatment.

**Author Contributions:** Conceptualization, J.L. and J.W.; methodology, J.L.; software, J.L.; formal analysis, J.L. and J.Z.; investigation, S.W., W.C. and X.W.; resources, J.W.; data curation, J.L.; writing—original draft preparation, J.L.; writing—review and editing, Z.S., X.M. and X.F.; visualization, J.L.; supervision, J.W. and X.F.; project administration, J.W.; funding acquisition, J.W. All authors have read and agreed to the published version of the manuscript.

**Funding:** This research was funded by the Shandong modern agricultural industry technical system project (SDAIT-20-04), the Key Research and Development Plan of Shandong Province (2017CXGC1302), the Shandong Province Natural Science Foundation (ZR2020QB169), and the National Key Research and Development Program (2017YFC1702700).

**Data Availability Statement:** Raw data are available from the corresponding author.

**Acknowledgments:** The authors would like to acknowledge the support provided by the State Key Laboratory of Crop Biology at Shandong Agricultural University.

**Conflicts of Interest:** The authors declare no conflict of interest.

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
