# Peer review of "Effects of Potassium Application on Growth and Root Metabolism of Salvia miltiorrhiza under Drought Stress"

_agronomy, doi:10.3390/agronomy13112796_

Round 1

Reviewer 1 Report

Comments and Suggestions for Authors

Dear Authors,

you have prepared an interesting text on the effect of potassium on growth and metabolism under drought conditions.

The paper has the character of a typical agronomic study, with well-presented results and an interesting discussion of the results. I would like you to refine a few areas, primarily to expand the introduction slightly and include some methodological comments. 

In the introduction section, I would add some information about the technology itself for growing Salvia miltiorrhiza sage in the area shown (China).

Please specify in the text what area in the sowing structure S. miltiorrhiza occupies in China. 

Was the experiment conducted in a vegetation hall? greenhouse? was there a controlled atmosphere?

It is useful to mark the study area on the map.

How long did the test last? what variety of sage was used? How many repetitions were used?

Please add 5 publications from the last 5 years on similar topics in the discussion of results.

Author Response

Dear Reviewer:

We appreciate for Editors and Reviewers’ warm work earnestly, and hope that the correction will meet with approval.

Once again, thank you very much for your comments and suggestions.

Sincerely,

Jianhua Wang

Reviewer 2 Report

Comments and Suggestions for Authors

Dear authors,

Congrats for the hard work you have done to obtain such a big amount of results. I can see that you have made a lot of analysis and measurements to collect all the raw data which were also statistically interpreted. 

Few recommendations were added to the pdf document attached.  

In the material and method section, please make sure that you have cited the author of the protocol you have used in the research or mention if it's an original one. 

The results are clearly presented in the paper and the discussions are consistent and relevant for the study as well as the cited references. 

Reviewer 3 Report

Comments and Suggestions for Authors

Dear Authors,

I enjoyed reading and reviewing this article which mainly focus on the potassium application and its effects on growth and root metabolism under drought conditions. I feel authors have put a great effort to support their research hypothesis but this article has several deficiencies in the experimental layout and the methodologies. I have highlighted all those deficiencies and the concerns directly in the attached manuscript draft. I highly encourage to address the highlighted issues before it can be considered for publication in the Agronomy.

Sincerely,

Comments on the Quality of English Language

Please check my comments in the draft and some sentences needs to be re-phrased for clear concept.

Author Response

(The authors gave the same response as above.)

Round 2

Reviewer 1 Report

Comments and Suggestions for Authors

I accept for publication. Thank you for all amendments.

Reviewer 3 Report

Comments and Suggestions for Authors

Dear Authors,

Thanks  for all the revisions and the manuscript looks way better now. I recommend publishing in Agronomy.

Regards